# Chinese music teachers value effectiveness, social, and facilitating factors over ease of use in ICT integration: A PLS-SEM study

Jun Jiang[1◉], Xiangming Zhang[2◉¤*]

1 Music College, Shanghai Normal University, Shanghai, China, 2 School of Arts, University of Hull, Hull, United Kingdom

◉ These authors contributed equally to this work.
¤ Current address: Department of Teacher Education, Norwegian University of Science and Technology (NTNU), Trondheim, Norway
* xiangming.zhang@ntnu.no

## Abstract

The integration of information and communication technology (ICT) in education has been widely acknowledged for its potential benefits. However, there is limited understanding of the factors influencing ICT integration among Chinese music teachers. This research explored technology acceptance among China's music teachers via the unified theory of acceptance and use of technology (UTAUT) to bridge existing knowledge gaps. A survey was conducted with 83 music teachers who used ICT in their teaching. Partial least squares structural equation modelling with effect size estimation revealed that Chinese music teachers' Behavioural Intention to use ICT was the strongest predictor of their actual Usage Behaviour. Performance Expectancy, Social Influence, and Facilitating Conditions positively influenced Behavioural Intention but did not directly impact Usage Behaviour, suggesting an indirect effect through shaping teachers' willingness to use ICT. Effort Expectancy had no effect on either Behavioural Intention or Usage Behaviour, though multigroup analysis revealed this relationship varied by teacher age and experience. This study demonstrates the applicability of the UTAUT model in understanding ICT integration among Chinese music teachers. The findings highlight the need to prioritise demonstrating benefits of ICT, creating supportive environments, and providing adequate resources and training for ICT in music education, with implementation strategies tailored to teachers' demographic characteristics. These findings suggest potential strategies for fostering technology integration in music education contexts.

## Introduction

Scholars have extensively studied the implementation of information and communication technology (ICT) in educational settings. While proponents argue that these

**Data availability statement:** Research data are available from https://doi.org/10.57760/sciencedb.08098.

**Funding:** The author(s) received no specific funding for this work.

**Competing interests:** The authors have declared that no competing interests exist.

technologies offer new possibilities for teaching and learning [1,2], critics caution against the uncritical adoption of these technologies and the potential commodification of education [3,4]. Despite global digitalisation efforts, the Organisation for Economic Co-operation and Development [5] reports that unequal access to quality ICT resources remains a significant challenge, emphasising that the effectiveness of ICT integration depends on incorporating these tools into pedagogical practices. In music education, debates also focus on balancing technological skills with the development of musicality and creativity. While some scholars have cautioned against an overemphasis on technology [6], others have explored how ICT can positively impact pedagogy and learning outcomes in music classrooms [7,8]. These discussions highlight the need for a deeper understanding of the role of technology in music education.

In this study, we view technology as a social process [9], encompassing not only physical equipment but also the interplay of pedagogical strategies, social interactions, and learning environments. Within music education, ICT tools are digital resources that facilitate musical experiences and learning, including composition software, digital audio workstations, and online collaborative platforms [10]. Therefore, the integration of ICT here refers to the process by which educators and learners accept, implement, and meaningfully incorporate these tools and practices into teaching and learning. This perspective differs from the broader view of educational technology as a field of research and practice [11], representing a focused subset within this broader field.

Research has shown that ICT effectively supports and enhances music teaching practices by assisting composition, fostering creativity, and developing students' listening, vocal expression, instrumental expression, and musical language skills [12,13]. Despite these potential benefits, a considerable proportion of music teachers do not integrate ICT into their classrooms, often due to limited resources and training [14,15]. The disparity between ICT capabilities and their real-world implementation highlights the importance of understanding music teachers' perceptions and attitudes towards technology integration. In China, the integration of ICT in music education presents unique challenges shaped by cultural, infrastructural, and pedagogical factors. While major urban areas have seen rapid technological advancement, a significant digital divide persists among different regions [16]. This disparity is further complicated by systemic challenges, including the hierarchical structure of the education system, disparities between urban and rural teacher expertise, a pervasive exam-centric culture, and significant parental influence on educational priorities [17–19]. A recent study has shown limited integration of digital technologies in music education at the university level, with 34% of students reporting no use of digital technologies in their studies [20].

To address this gap and provide a theoretical framework for examining music teachers' acceptance and use of ICT in China, this research was guided by the unified theory of acceptance and use of technology (UTAUT) model. The framework comprises four core constructs: Performance Expectancy (PE); Effort Expectancy (EE); Social Influence (SI); and Facilitating Conditions (FC) [21]. This model has been successfully applied in various educational contexts using different technologies, including e-learning websites [22], learning management software [23], and ICT

tools [24]. Recent research has extended the application of UTAUT-based models to examine ICT integration in higher education teachers' research processes, finding that factors such as digital skills, ethics, digital flow, and behavioural intention significantly influence ICT adoption [25]. However, most UTAUT research has focused on education in Asia and North America [26], with limited attention to music education in China, highlighting a significant gap in the literature.

The current study examines the extent to which UTAUT constructs predict Chinese music teachers' intentions to use ICT and their actual usage behaviours. By analysing this under-researched sample, we attempt to evaluate the effectiveness of the UTAUT model in a specific educational setting in China. Understanding these factors may be crucial for improving creativity, technological skills, and pedagogical engagement in music education, bridging the gap between ICT's potential and its implementation in Chinese music classrooms. The research seeks to inform equitable use policies, guide effective teacher training programs, and advance the discussion on the role of technology in music education in China.

## Literature review

Integrating a variety of technologies into music education, including digital audio workstations, interactive learning platforms, and online collaboration tools, has created opportunities for innovative and potentially transformative learning experiences [27,28]. However, music educators have exhibited varied responses to these technological advancements. Some teachers are reluctant to incorporate technology into their practices due to a lack of individual beliefs about technological adoption [29]. Others prioritise practical, applied teaching methods such as vocal and instrumental instruction, adhering to the master-apprentice model [30].

While ICT integration in music education has gained global attention, research on Chinese music teachers' attitudes towards this technological shift remains limited. Studies have explored the potential of various music technologies in educational settings, such as Open Sound Control-based interactive music systems [31,32]. The few studies examining Chinese music teachers' perspectives [29] suggest that their individual beliefs interact with practical challenges. These works indicate that a range of factors influence teacher attitudes, including technological acceptance, technology literacy, and cultural considerations specific to Chinese music education.

This research gap becomes more apparent given the speed of technological change and how attitudes may shift over time. Understanding the factors impacting how music teachers in China incorporate and adopt ICT could be crucial for developing effective strategies to enhance music education in the digital age.

## The unified theory of acceptance and use of technology (UTAUT)

Given the complexity of technology adoption in educational settings, a comprehensive theoretical framework is necessary to examine the multiple factors at play. UTAUT integrates eight well-known models and theories, including the technology acceptance model. This integration demonstrates higher explanatory power (up to 70% variation) than the technology acceptance model alone in predicting users' technology use intentions and behaviours [21]. UTAUT suggests that Behavioural Intention (BI) and Use Behaviour (UB) are influenced by four core factors (see Fig 1). PE is the extent to which users perceive that the technology improves their job performance. EE refers to perceived ease of use. SI represents the extent to which users believe that others may influence their use of the technology. FC refers to users' beliefs about the availability of organisational and technical infrastructure that support technology use. According to the model, PE, EE, and SI predict BI, which in turn predicts UB. Additionally, FC is hypothesised to directly influence UB. Gender, age, experience, and voluntariness of use were identified as moderators in the original UTAUT model affecting BI or UB relationships.

## Application of UTAUT in educational contexts

Initially applied in information systems research, UTAUT has been widely adopted and adapted in educational contexts. Wan et al. [33] used it to study MOOC adoption among Chinese university students, while Guillén-Gámez et al. [34] extended it to examine teachers' use of YouTube for education. These applications demonstrate UTAUT's flexibility in

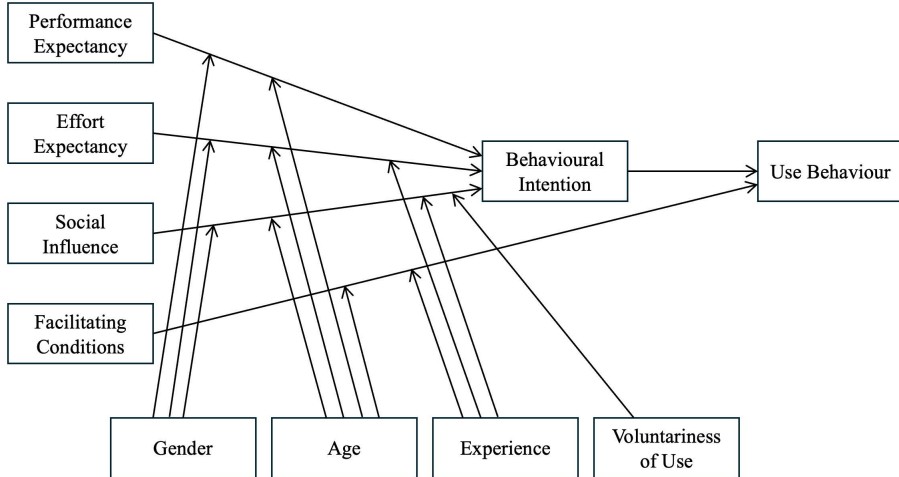

**Fig 1. The UTAUT model.** *Adapted from* "User acceptance of information technology: Toward a unified view," by V. Venkatesh, M. G. Morris, G. B. Davis, and F. D. Davis, 2003, *MIS Quarterly, 27*(3), p. 447 (https://doi.org/10.2307/30036540). Copyright 2003 MIS Quarterly.

studying various educational technologies. Numerous studies have confirmed that BI positively predicts UB [35–37]. However, empirical evidence on how other constructs impact technology adoption remains inconsistent.

Several studies have investigated the role of PE in technology adoption in education and have found mixed results. While some researchers have found that PE positively affects BI [38,39] and UB [40,41], others found no significant association between PE and technology acceptance [24,42,43]. These mixed results suggest that expected technological benefits may poorly forecast actual implementation in academic settings. Similarly, the influence of EE and SI on technology acceptance varies across studies. Some researchers have found that EE and SI positively affect BI [44,45] and UB [36,42], indicating that perceived ease of use and social pressures can drive technology adoption. However, other studies have reported non-significant relationships between EE and SI and technology acceptance [40,46,47], suggesting that these factors may not always be influential. Some researchers have found that FC positively predicts both UB [48,49] and BI [37,50], suggesting that the availability of resources and support can directly influence both the intention to use and actual usage of technology. However, other studies have shown non-significant effects of FC on UB [51] and BI [52], indicating that facilitating conditions may not always translate into increased technology adoption.

Regarding the moderators, the moderating effects proposed in the original UTAUT model have yielded mixed results. While some studies discuss that gender has been found to moderate the effects of PE, EE, and SI on BI, with females often being more influenced by social factors [53], researchers did not find gender to significantly moderate PE, EE, or SI on the actual adoption and use of electronic instructional media. Age was found to be a significant direct moderator of behavioural intention in one study, indicating that preservice teachers' intention to use ICT decreased as their age increased [54]. Conversely, other research reported no significant moderating effects for gender, age, or experience on the relationships between PE, EE, SI, and BI at all [55]. Additionally, voluntariness of use was not found to be a significant moderator of BI in one empirical study, despite participants largely perceiving ICT use as voluntary [54]. These inconsistent findings across different educational contexts underscore the complexity of technology adoption in education and suggest the need for further testing of these moderators in specific educational domains such as music education.

Despite the widespread use of UTAUT in educational research, its application in music education settings has been limited. While some scholars have explored broader technology use and integration by music teachers [29,56], specific research on ICT integration by Chinese music teachers remains lacking, highlighting a significant gap in the literature. These gaps, combined with inconsistencies in findings across educational contexts, underscore the need for further

investigation. By examining the factors influencing Chinese music teachers' integration of ICT through UTAUT, this study aims to contribute to a more comprehensive understanding of technology acceptance in diverse educational and cultural contexts. The findings may help to refine the UTAUT model and inform strategies for promoting ICT integration in music education.

## Hypothesis development

In this study, we used the UTAUT model to investigate Chinese music teachers' engagement with ICT. We predicted that PE, EE, SI, and FC would influence BI and UB. Specifically, teachers who believe that ICT will improve their teaching effectiveness (PE) and perceive ICT as easy to use (EE) are more likely to intend to use and actually use these technologies. In addition, teachers who perceive pressure from important others such as colleagues and administrators to use ICT (SI) and believe they have the necessary resources and support (FC) are more likely to adopt and integrate these tools into their teaching.

We also examined the moderating effects of gender, age, and experience, as specified in the original UTAUT model [21]. Investigating these moderators allowed for a more robust and informative model with practical implications for implementation strategies. However, voluntariness of use was not included because all participants had prior ICT experience, ensuring a comparable baseline of voluntary engagement.

Fig 2 illustrates the proposed variable relationships. The study seeks to validate the associations between PE, EE, SI, FC, BI, and UB among Chinese music teachers' ICT integration. The specific hypotheses are as follows:

- Hypothesis 1 (H1): PE would positively predict BI

- Hypothesis 2 (H2): PE would positively predict UB

- Hypothesis 3 (H3): EE would positively predict BI

- Hypothesis 4 (H4): EE would positively predict UB

- Hypothesis 5 (H5): SI would positively predict BI

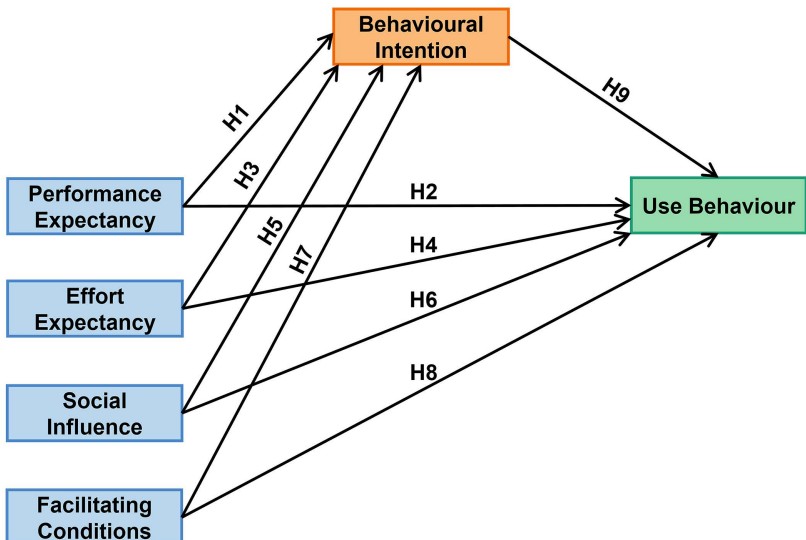

**Fig 2. Proposed partial mediation model and the hypotheses.**

- Hypothesis 6 (H6): SI would positively predict UB

- Hypothesis 7 (H7): FC would positively predict BI

- Hypothesis 8 (H8): FC would positively predict UB

- Hypothesis 9 (H9): BI would positively predict UB

## Method

### Sample

In alignment with prior research on UTAUT, we employed homogeneous convenience sampling [57], recruiting accessible individuals who share key characteristics relevant to the study. Specifically, we targeted music teachers in Chinese higher education institutions, reducing heterogeneity compared to including teachers from primary or secondary schools and ensuring comparable institutional contexts, teaching environments, and professional requirements. Homogeneous convenience samples are more representative, less biased, and more generalisable than conventional convenience samples that mix diverse populations [57].

Sample size was determined following guidelines for partial least squares-based structural equation modelling (PLS-SEM). Using the inverse square root method [58–60] implemented in WarpPLS (Version 8.0) [59,61], the recommended minimum sample size was 69 to detect a minimum absolute significant path coefficient of .30 at the .05 significance level with 80% power. This method is conservative and slightly overestimates the sample size needed [62]. Moreover, the overall complexity of a structural model has little influence on sample size requirements for PLS-SEM [62].

This study was approved by the Ethics Committee of the University of Hull (No. 1920PGR01) and conducted in line with the Declaration of Helsinki. Recruitment took place from 01/10/2019 to 31/12/2019. Of 144 voluntary participants, 61 were excluded due to not using ICT in their teaching practice, submitting invalid questionnaires with uniform responses, having more than 10% missing data, or completing the survey unreasonably quickly [63,64]. The final analysis included 83 teachers (32 males, 51 females) from 21 provinces. Gender distribution (38.6% males, 61.4% females) aligns with previous reports on Chinese music teachers [65]. Most participants (80.7%) were aged 24–44 years, and 81.9% had over 3 years of instructional experience.

Our final sample size ($n = 83$) exceeds the minimum requirement for detecting medium-sized effects and is adequate for the model tested. Also, it is comparable to sample sizes in other exploratory technology adoption studies (ranging from $n = 85$ to $n = 95$) [66,67]. Despite the strict inclusion criteria, the geographic diversity of 21 provinces enhances its representativeness within the target population of Chinese higher education music teachers.

### Measures and procedure

We used a UTAUT-based instrument to measure the level of ICT acceptance among Chinese music educators. This scale consisted of six subscales measuring six latent constructs: PE; EE; SI; FC; BI; and UB. The items for the former five constructs were adapted from those in the UTAUT model [21], which is commonly used to assess individuals' or organisations' behavioural intentions to accept and use technology. The items for the last construct were adapted from previous studies [68,69]. Although these items have been validated in prior research, we conducted reliability and validity assessments within our sample to ensure appropriate psychometric properties, as detailed in the results section. Overall, the scale comprises 22 items with four items each for PE, EE, SI, and FC and three items each for FC and UB (see Table 1). Items were scored on a 7-point Likert scale (1 = *strongly disagree*, 7 = *strongly agree*), where higher values reflected greater construct levels.

A questionnaire comprising informed consent, demographic items, and key scales was developed and disseminated via a web-based survey platform (www.wenjuan.com). Invitations containing the survey link were disseminated

**Table 1. Scale items.**

| Subscale | Item |
|---|---|
| **Performance Expectancy (PE)** | PE1: I would find ICT useful in my job. |
| | PE2: Using ICT enables me to accomplish tasks more quickly. |
| | PE3: Using ICT increases my productivity. |
| | PE4: If I use ICT, I will increase my chances of becoming more competent in teaching. |
| **Effort Expectancy (EE)** | EE1: My interaction with ICT would be clear and understandable. |
| | EE2: It would be easy for me to become skilful at using ICI. |
| | EE3: I would find ICT is easy to use. |
| | EE4: Learning to use ICT is easy for me. |
| **Social Influence (SI)** | SI1: People who influence my behaviour think that I should use ICT. |
| | SI2: People who are important to me think that I should use ICT. |
| | SI3: I use ICT because of the proportion of co-workers who use it. |
| | SI4: In general, the administration of my organisation has supported the use of ICT. |
| **Facilitating Conditions (FC)** | FC1: I have the resources necessary to use ICT. |
| | FC2: I have the knowledge necessary to use ICT. |
| | FC3: ICT is compatible with other applications (teaching methods) I use. |
| | FC4: A specific person (or group) would be available for assistance with difficulties when using ICT. |
| **Behavioural Intention (BI)** | BI1: I intend to use ICT in the future. |
| | BI2: I predict I would use ICT in the next <n> months. |
| | BI3: I would like to further recommend to other teachers to use ICT. |
| **Use Behaviour (UB)** | UB1: I use ICT on a regular basis. |
| | UB2: I use ICT frequently. |
| | UB3: I use a lot of tools of ICT. |

*Note*. ICT = information and communication technology.

through email, social media, and professional networks. Automatic reminders were sent after one week to non-respondents. Only individuals who provided consent were allowed to proceed with the subsequent steps. Participants began by providing their demographic information, followed by completing the six subscales in sequence, typically finishing within 15 minutes.

## Data analysis

The proposed model and hypotheses were tested using PLS-SEM. Typically, there are two types of SEM [62,70]: covariance-based structural equation modelling (CB-SEM) and PLS-SEM. PLS-SEM was selected for several reasons [62,71–73]. First, the exploratory nature of this study, with its primary focus on predicting and explaining a target construct (i.e., UB), made PLS-SEM appropriate. Second, CB-SEM requires normally distributed data, whereas PLS-SEM operates without distributional constraints, better accommodating Likert-scaled responses that may violate normality assumptions. Third, PLS-SEM seeks to explain all observed measurement variance, which can be advantageous in certain research contexts [74]. In contrast, CB-SEM focuses on common variance, excluding random error variance and measure-specific variance components from the measurement model [75].

PLS-SEM was conducted using SmartPLS (Version 4.1.1.4) [76]. The analysis followed a two-stage approach [72,77]: measurement model reliability and discriminant validity were first assessed using the PLS algorithm with 1,000 iterations,

followed by structural model assessment to examine path coefficients between latent variables [78]. Also, this study emphasised effect sizes (ESs) and 95% confidence intervals (CIs) over *p*-values for result interpretation. While *p*-values indicate statistical significance, they do not assess the magnitude or practical significance of an effect [79,80]. ESs and CIs quantify the strength of effects and the precision of estimates, becoming key metrics in statistical studies [81–83]. Current statistical practice emphasises ESs and CIs over *p*-values for result interpretation [84–86]. A significant effect is indicated when the CI of an ES does not contain zero, and no effect is indicated when it does [87–89]. Standardised coefficients (*β*) of .10, .30, and .50 represent small, medium, and large effects, respectively [90].

## Results

### Measurement model assessment

Table 2 shows construct reliability and validity outcomes. Cronbach's α (.78–.92) and composite reliability (CR) (.86–.95) both exceeded the threshold of .70 [62], confirming adequate internal consistency across all measures. Convergent

**Table 2. Psychometric properties for the subscales.**

| Subscale and Item | *M* | *SD* | Factor loading | Cronbach's α | CR | AVE | VIF |
|---|---|---|---|---|---|---|---|
| **Performance Expectancy (PE)** | | | | | | | |
| PE1 | 5.94 | 1.15 | .83 | .88 | .92 | .74 | 1.95 |
| PE2 | 5.46 | 1.48 | .92 | | | | 3.53 |
| PE3 | 5.52 | 1.46 | .89 | | | | 3.04 |
| PE4 | 5.53 | 1.46 | .80 | | | | 1.89 |
| **Effort Expectancy (EE)** | | | | | | | |
| EE1 | 4.84 | 1.47 | .81 | .88 | .91 | .73 | 1.52 |
| EE2 | 4.57 | 1.50 | .86 | | | | 2.71 |
| EE3 | 4.70 | 1.55 | .89 | | | | 3.91 |
| EE4 | 4.66 | 1.45 | .86 | | | | 3.06 |
| **Social Influence (SI)** | | | | | | | |
| SI1 | 5.06 | 1.67 | .87 | .86 | .91 | .71 | 2.92 |
| SI2 | 4.98 | 1.65 | .87 | | | | 3.11 |
| SI3 | 4.12 | 1.74 | .85 | | | | 2.50 |
| SI4 | 4.93 | 1.68 | .78 | | | | 2.14 |
| **Facilitating Conditions (FC)** | | | | | | | |
| FC1 | 5.71 | 1.26 | .79 | .78 | .86 | .60 | 1.75 |
| FC2 | 5.01 | 1.36 | .82 | | | | 1.86 |
| FC3 | 5.93 | 1.22 | .77 | | | | 1.35 |
| FC4 | 3.87 | 1.92 | .73 | | | | 1.43 |
| **Behavioural Intention (BI)** | | | | | | | |
| BI1 | 5.41 | 1.61 | .93 | .89 | .93 | .82 | 3.28 |
| BI2 | 4.84 | 1.76 | .86 | | | | 2.06 |
| BI3 | 5.05 | 1.61 | .92 | | | | 2.99 |
| **Use Behaviour (UB)** | | | | | | | |
| UB1 | 5.07 | 1.57 | .92 | .92 | .95 | .87 | 3.09 |
| UB2 | 4.77 | 1.80 | .95 | | | | 4.34 |
| UB3 | 4.63 | 1.76 | .93 | | | | 3.61 |

*Note.* CR = composite reliability; AVE = average variance extracted; VIF = variance inflation factor.

validity assessment used factor loadings and average variance extracted (AVE). All items demonstrated loadings between .73 and .95, exceeding the recommended .70 threshold. Furthermore, AVE values for all constructs ranged from .60 to .87, surpassing the minimum threshold of .50 suggested by Hair et al. [62]. These findings confirm that the items consistently measure the same concept.

We assessed discriminant validity using the Fornell-Larcker criterion [91]. Table 3 shows diagonal values (bold) exceed off-diagonal correlations in respective rows and columns, confirming that constructs relate more strongly to their own indicators than to others. The Heterotrait-Monotrait (HTMT) criterion provided additional discriminant validity assessment, recognised as more rigorous [92,93]. Table 4 displays HTMT results from bootstrap analysis (5,000 samples, 95% confidence). Nearly all HTMT values remained below .90, confirming construct discriminant validity [62]. The BI-UB HTMT value (.94) slightly exceeded this threshold but remains acceptable, reflecting the theoretically expected strong relationship between behavioural intentions and actual usage [21,94].

## Structural model assessment

Multicollinearity assessment preceded structural model evaluation through variance inflation factor (VIF) analysis. Table 2 shows that the VIF values (1.35–4.34) are lower than the threshold of 5 recommended by Hair et al. [95], confirming the absence of multicollinearity issues. Fig 3 presents PLS-SEM outcomes, displaying path coefficients and 95% CIs, $t$, and $p$ values. Structural analysis confirmed four hypotheses (H1, H5, H7, H9) while rejecting five others (H2, H3, H4, H6, H8) in the overall model. The coefficient of determination ($R^2$) values for BI and UB were .72 and .77, respectively. According to Hair et al. [62], these $R^2$ values demonstrate substantial explanatory power. The model accounts for 72% of variance in music educators' ICT behavioural intentions through PE, EE, SI, and FC perceptions. Additionally, behavioural intentions explain 77% of variance in teachers' actual ICT usage.

**Table 3. Discriminant validity analysis (Fornell-Larcker).**

| Variable | 1 | 2 | 3 | 4 | 5 | 6 |
|---|---|---|---|---|---|---|
| 1. Performance Expectancy | **.86** | | | | | |
| 2. Effort Expectancy | .63 | **.85** | | | | |
| 3. Social Influence | .67 | .58 | **.84** | | | |
| 4. Facilitating Conditions | .56 | .53 | .68 | **.78** | | |
| 5. Behavioural Intention | .72 | .59 | .76 | .73 | **.90** | |
| 6. Use Behaviour | .60 | .55 | .76 | .74 | .85 | **.93** |

*Note.* Bold numbers on the diagonal denote the square roots of AVEs.

**Table 4. Discriminant validity analysis (HTMT).**

| Variable | AVE | 1 | 2 | 3 | 4 | 5 | 6 |
|---|---|---|---|---|---|---|---|
| 1. Performance Expectancy | .74 | — | | | | | |
| 2. Effort Expectancy | .73 | .68 | — | | | | |
| 3. Social Influence | .71 | .76 | .63 | — | | | |
| 4. Facilitating Conditions | .60 | .65 | .59 | .81 | — | | |
| 5. Behavioural Intention | .82 | .80 | .64 | .87 | .85 | — | |
| 6. Use Behaviour | .87 | .66 | .58 | .85 | .86 | .94 | — |

*Note.* AVE = average variance extracted.

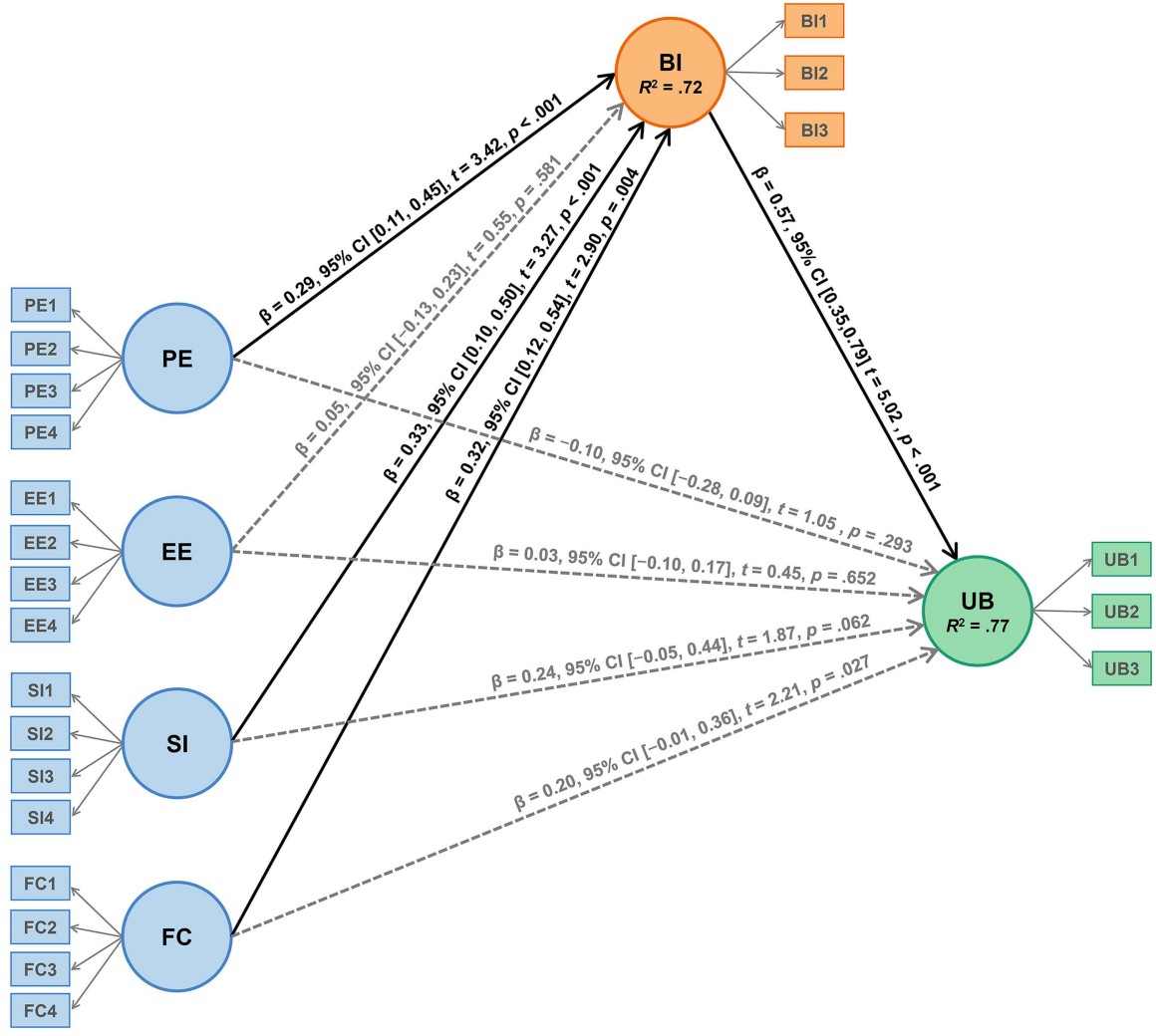

**Fig 3. The final structural model.** *Note.* PE = Performance Expectancy; EE = Effort Expectancy; SI = Social Influence; FC = Facilitating Conditions; BI = Behavioural Intention; UB = Use Behaviour; CI = Confidence Intervals. The bias-corrected CIs, resulting from the implementation of the bias-corrected and accelerated bootstrap for the dataset calculations, are reported here.

Beyond assessing model fit, we evaluated the model's out-of-sample predictive power using the PLSpredict procedure [72]. We used 10-fold cross-validation with 10 repetitions to evaluate prediction accuracy. Results indicated strong predictive capability for the endogenous constructs, with $Q^2_{predict}$ values of .66 for BI and .63 for UB. Benchmark comparisons showed that all construct indicators had lower root mean square error and mean absolute error than the linear regression benchmark, confirming high predictive power and substantial predictive utility beyond simpler linear models [96].

## Multigroup analysis

To examine the potential moderating effects of gender, age, and experience, we conducted a multigroup analysis (MGA) using a nonparametric permutation test with 1,000 permutations [97]. The sample was divided into subgroups for moderation analysis based on three variables: gender (female, $n = 51$; male, $n = 32$), age (younger, 24–34 years, $n = 31$; older, ≥ 35

years, $n = 52$), and experience (less experienced, ≤ 10 years, $n = 43$; more experienced, > 10 years, $n = 40$). Table 5 reports the group-specific path coefficients, the between-group differences, and the corresponding $p$-values.

As shown in Table 5, no gender differences were observed across the paths. In contrast, age and teaching experience moderated several relationships. Specifically, the effects of SI on BI and EE on UB were stronger among younger than older teachers, and also stronger among less experienced than more experienced teachers. Conversely, the effect of FC on BI was stronger among experienced teachers. These results suggest that gender does not moderate the relationships, whereas age and teaching experience do.

Additionally, model explanatory power varied substantially across demographic groups compared to the overall model (BI: $R^2 = .72$, UB: $R^2 = .77$). Male teachers demonstrated the highest model performance (BI: $R^2 = .78$, UB: $R^2 = .88$), while female teachers showed moderate performance (BI: $R^2 = .68$, UB: $R^2 = .71$). Similarly, younger teachers (BI: $R^2 = .81$, UB: $R^2 = .82$) and less experienced teachers (BI: $R^2 = .83$, UB: $R^2 = .79$) exhibited higher explained variance compared to their older and more experienced counterparts. These patterns indicate differential model effectiveness and underscore the importance of demographic-specific approaches to technology adoption and implementation.

## Discussion

This study investigated factors influencing Chinese music teachers' adoption and integration of ICT in their teaching practices. Key findings revealed that BI was positively affected by PE, SI, and FC, but not by EE in the overall model. UB was only directly influenced by BI, suggesting these factors operate through teachers' intentions rather than directly influencing behaviour. Importantly, multi-group analysis revealed that these relationships varied significantly across demographic groups, particularly for EE effects among different age and experience levels.

### Impact of performance expectancy on usage behaviour

The first key finding is the indirect influence of PE on UB through BI, which supports hypotheses H1 and H9, but not H2. This finding is consistent with previous research that has demonstrated a positive impact of performance expectancy on usage behaviour [30]. Our results extend these findings by showing an indirect effect mediated by BI. This outcome suggests that when music teachers perceive ICT as beneficial for enhancing their teaching performance, they develop stronger intentions to use it, which subsequently drives actual usage. However, our results contrast with those of Teo and Noyes [98], who found a direct effect of PE on UB among pre-service teachers. This difference may be attributed to several factors, including the characteristics of our sample, or the ICT tools considered in each study. Further research is

**Table 5. Results of multigroup analysis.**

| Path | Gender | | | | Age | | | | Experience | | | |
|---|---|---|---|---|---|---|---|---|---|---|---|---|
| | Female | Male | Difference | $p$ | Younger | Older | Difference | $p$ | Less | More | Difference | $p$ |
| PE→BI | .33 | .19 | .14 | .472 | .27 | .25 | .02 | .927 | .25 | .31 | −.06 | .758 |
| PE→UB | .04 | −.25 | .29 | .183 | −.28 | −.06 | −.22 | .304 | −.14 | −.06 | −.08 | .720 |
| EE→BI | .02 | .14 | −.12 | .580 | −.01 | .06 | −.06 | .790 | .05 | −.04 | .10 | .674 |
| EE→UB | −.01 | .08 | −.09 | .586 | .39 | −.05 | .44 | **.006** | .23 | −.04 | .27 | **.072** |
| SI→BI | .36 | .26 | .10 | .691 | .54 | .16 | .38 | **.088** | .52 | .07 | .45 | **.035** |
| SI→UB | .25 | .19 | .06 | .834 | .03 | .31 | −.28 | .378 | −.03 | .34 | −.37 | .175 |
| FC→BI | .25 | .40 | −.15 | .552 | .21 | .48 | −.27 | .311 | .18 | .58 | −.40 | **.095** |
| FC→UB | .24 | .12 | .13 | .550 | .03 | .20 | −.16 | .450 | .16 | .20 | −.04 | .844 |
| BI→UB | .42 | .80 | −.38 | .153 | .73 | .54 | .19 | .470 | .71 | .52 | .19 | .463 |

*Note.* Moderation effects were considered present when $p < .10$, with such results highlighted in bold. PE = Performance Expectancy; EE = Effort Expectancy; SI = Social Influence; FC = Facilitating Conditions; BI = Behavioural Intention; UB = Use Behaviour.

needed to understand the exact reasons for this discrepancy. Notably, the strong influence of PE was consistent across teachers of different ages, genders, and experience levels. This suggests that perceived ICT usefulness is a universal driver for all music teachers in this context.

Based on these findings, we propose that efforts to promote ICT integration among music teachers could emphasise the specific ways in which technology can enhance music instruction and learning outcomes. Professional development programmes could focus on demonstrating practical applications of ICT in music classrooms, potentially increasing teachers' performance expectancy and their intention to use and actual usage of ICT.

### Impact of social influence on usage behaviour

Our second key finding is that SI indirectly impacts UB through BI, supporting hypotheses H5 and H9 but not H6. This result aligns with previous research in general education [36] and the original UTAUT model [21]. The indirect effect of SI on UB through BI underscores the importance of social and professional networks in shaping technology adoption behaviours. This finding is consistent with a study on MOOC continuance intention conducted by Wu and Chen [99] in China, which also found an indirect relationship between SI and continuance intention to use. These findings suggest that fostering a supportive professional environment may be crucial for promoting ICT integration among music teachers. This could involve creating opportunities for peer learning, highlighting successful ICT integration practices, and cultivating a school culture that values technological innovation in music education.

However, our MGA results revealed a more nuanced picture of SI. This factor was significantly more powerful in driving BI for less experienced and younger teachers. This suggests that early-career educators may be more sensitive to cues from their social environment, such as colleagues and administrators, when deciding whether to adopt new technology. In contrast, veteran teachers appear to rely more on their own judgment, with SI playing a minimal role. This has direct implications for implementation strategies, indicating that fostering peer-to-peer support and leadership endorsement is particularly effective for engaging novice teachers.

### Impact of facilitating conditions on usage behaviour

Our third finding is that FC indirectly impacts UB through BI, supporting H7 and H9 but not H8. This outcome is consistent with previous research suggesting FC can positively affect UB [14,21]. Our study extends these findings by identifying an indirect effect mediated by BI. This result suggests that the availability of resources and support influences teachers' intentions to use ICT, which in turn affects their actual usage. This finding also aligns with research conducted by Wut et al., [100] on student-to-student interaction within online learning platforms, where they found a relationship between FC and UB mediated by other factors. Given these findings, educational institutions may consider investing in robust technological infrastructure and ongoing professional development opportunities for music teachers. These efforts could enhance teachers' perceptions of facilitating conditions, potentially increasing their intentions to use ICT and their actual usage.

Interestingly, the importance of FC varied by teacher experience. While it had a modest effect on the intentions of less experienced teachers, it was a strong predictor for their more experienced colleagues, suggesting that veteran educators, through years of practice, are more pragmatic and aware of institutional barriers. Their intention to use ICT is therefore heavily contingent on tangible evidence of support, such as available resources, technical assistance, and training. This finding implies that to successfully implement new technologies among senior staff, school leaders could prioritise and clearly communicate the availability of robust FC.

### Non-significant impact of effort expectancy on usage behaviour

A notable finding of our study is the lack of a significant effect of EE on UB, which contradicts H3 and H4. This result is inconsistent with previous studies that have shown a positive relationship between EE and UB [14,15], but aligns with studies in specialised fields in China. For instance, Pan and Gao [101] found that EE did not significantly affect BI among

nurses adopting a mobile nursing app, suggesting that in specialised domains, including music education, factors other than ease of use may be more critical in technology adoption decisions.

However, the MGA results revealed that the influence of EE on music teachers' UB differs significantly by experience level. EE showed significant positive effects for younger and less experienced teachers, while this effect disappeared for older and more experienced educators. This finding supports the original UTAUT proposition that EE influence diminishes as users gain experience [21], highlighting the importance of considering demographic moderators in technology acceptance research.

### Limitations and future directions

While this study offers valuable insights into ICT integration among Chinese music teachers, several limitations should be acknowledged. First, the sample size, while methodologically suitable for PLS-SEM analysis and comparable to similar specialised studies, limits the statistical generalisability of our findings. Furthermore, we used a homogeneous convenience sample of music teachers from Chinese higher education institutions who actively use ICT. Although this specificity was beneficial for reducing confounding variables and ensuring relevance, it limits the applicability of our findings to other populations, such as music educators in K-12 settings or different cultural contexts. While the convenience sampling approach achieved geographic diversity across 21 provinces, it may not fully represent all teachers in higher education in China, particularly those less inclined to participate in technology-related research. Future studies should aim to validate these findings with larger, more diverse, and randomly selected samples.

Second, while this study conducted an MGA to explore moderating effects, the subgroup sizes for age and gender analyses were relatively small, limiting the statistical power to detect smaller-sized effects. Thus, these findings deserve careful consideration. Future research with larger samples is encouraged to corroborate these results.

Third, our study did not specify the types of ICT tools examined. Different ICT tools may exhibit varying levels of acceptance and usage patterns among music teachers. Future research could identify and categorise these tools to provide more nuanced insights into technology adoption in music education.

Lastly, although the UTAUT model served as a useful framework for our study, it may not encompass all factors relevant to ICT adoption in music education. Future research could explore additional variables, such as teachers' musical backgrounds or specific pedagogical beliefs, that might influence ICT adoption in this unique context.

### Conclusion

This research examined factors affecting ICT adoption among Chinese music educators through the UTAUT model. Results demonstrated that PE, SI, and FC indirectly affected UB through BI, whereas EE showed no significant impact on either intention or behaviour in the overall model. However, the influence of these factors varied across demographic groups, particularly by age and experience.

Theoretically, this research validates and extends UTAUT within Chinese music education contexts, providing empirical evidence for behavioural intention's mediating role between key constructs and usage behaviour. These insights enhance understanding of ICT integration processes among music educators and inform future technology acceptance research in specialised educational domains. Practically, the findings offer important implications for educational stakeholders, including educational institutions, policymakers, and music teacher training programmes. Efforts to promote ICT integration among music teachers could focus on three key areas: (1) emphasising the benefits of ICT in enhancing teaching performance and student learning outcomes; (2) fostering a supportive social environment that values technology use in music education; and (3) providing adequate resources, support, and training for music teachers to develop their technological competence. Implementation strategies could include developing targeted professional development programmes, establishing peer-mentoring networks, and ensuring access to appropriate technological resources and support systems. Schools could develop assessment tools based on the identified factors to evaluate and continuously improve ICT

adoption strategies in music education. Our MGA results further suggest that the relative importance of these factors differs across demographic groups, indicating that implementation strategies may be more effective if tailored to teachers' specific age and professional experience.

In summary, this study provides empirical evidence on factors influencing music teachers' integration of ICT in China within the UTAUT model. These findings suggest potential strategies for technology integration that warrant further investigation, providing initial insights for ICT implementation strategies in similar contexts, though broader validation is needed before widespread application.

## Author contributions

**Conceptualization:** Xiangming Zhang.

**Data curation:** Xiangming Zhang.

**Formal analysis:** Jun Jiang, Xiangming Zhang.

**Investigation:** Xiangming Zhang.

**Methodology:** Jun Jiang, Xiangming Zhang.

**Project administration:** Xiangming Zhang.

**Resources:** Xiangming Zhang.

**Visualization:** Jun Jiang, Xiangming Zhang.

**Writing – original draft:** Jun Jiang, Xiangming Zhang.

**Writing – review & editing:** Jun Jiang, Xiangming Zhang.

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
