## [Decision Letter · Decision Letter 0]

27 Aug 2025

Dear Dr. Zhang,

We look forward to receiving your revised manuscript.

Kind regards,

Musa Adekunle Ayanwale, Ph.D

Academic Editor

PLOS ONE

Journal Requirements:

Reviewers' comments:

Reviewer's Responses to Questions

**Comments to the Author**

1. Is the manuscript technically sound, and do the data support the conclusions?

Reviewer #1: Yes

Reviewer #2: Yes

2. Has the statistical analysis been performed appropriately and rigorously?

Reviewer #1: Yes

Reviewer #2: Yes

3. Have the authors made all data underlying the findings in their manuscript fully available?

Reviewer #1: Yes

Reviewer #2: Yes

4. Is the manuscript presented in an intelligible fashion and written in standard English?

Reviewer #1: Yes

Reviewer #2: Yes

Reviewer #1: Giving the implementation of artificial intelligence in every area of human endeavor, the study is providing an important insight into how integration of ICT into music teaching and learning or it other application is very important. Meanwhile, the following comment could improve the standard of the study.

Minor comment

The authors should use the full meaning of acronyms in the first appearance such as OECD.

Use the full meaning of an acronyms for a subtopic such as at 2.1

What is homogeneous convenience sampling?

I was initially confused when the authors mentioned that the scales are already validated but later saw that validity evidences where provided as well in the paper. It will be useful to hint the reader that despite that the scales have been validated in previous studies, it will also be validated in the study (page 5).

It could be more convenient for readers to have the diagrams close to where they are discussed instead of having them at the appendix.

Major comment

A major confusion I have in the study is about the group sampled. At some point teachers in higher education institution was mentioned and at another point in-service teachers were mentioned. Are they the same?

I am worried about the sample size. Only 83 samples was used for the final analysis for such a complex model.

There are no discussion regarding the fit of the model. I think that is an important aspect of a model like this

Does the authors check if there are connections between the items or simply assumed there is none?

Why not run the same model with the available moderators and compare the fit of the model for the sample in the study. That could present a robust and more informative model that could inform policy.

Reviewer #2: The manuscript was written in a clear and coherent manner, and all the necessary aspects of the study were elucidated concisely. However, there is only one statement that needs the authors to adjust so that it is accurate and meaningful. The authors should add the phrase "and learning" to the end of the third sentence in paragraph 2, under the "Introduction" heading on page 1.

**Do you want your identity to be public for this peer review?** For information about this choice, including consent withdrawal, please see our Privacy Policy

Reviewer #1: No

Reviewer #2: No

---

## [Author Response · Author response to Decision Letter 1]

16 Sep 2025

Response to Academic Editor and Reviewers

Academic Editor

Based on the reviewer's concerns and my overall evaluation of the paper, the following points are noted: First, there is inconsistency in describing the target population, with unclear references to both “in-service teachers” and “teachers in higher education institutions,” creating confusion about the study context. Second, the sample size is inadequate for the complexity of the structural model, raising concerns about statistical power and generalisability. Third, the manuscript lacks a discussion on model fit indices, which are essential for evaluating SEM models. There is also no evidence that item correlations were assessed, which is a key step in validating construct measurement. Additionally, the study would benefit from including moderator analyses to strengthen its explanatory power and policy relevance. Finally, reference formatting deviates from APA style, and diagrams placed in the appendix would be more effective if integrated within the main text. These issues collectively warrant substantial revision to improve the clarity, methodological rigour, and overall presentation of the manuscript.

Response: Thank you for your comment. For most of the concerns you raised, please refer to our responses to Reviewer #1. Regarding the reference formatting issues, we have revised the manuscript to comply with PLOS ONE’s “Vancouver” citation style.

Reviewer #1

Giving the implementation of artificial intelligence in every area of human endeavor, the study is providing an important insight into how integration of ICT into music teaching and learning or it other application is very important. Meanwhile, the following comment could improve the standard of the study.

Minor Comments

The authors should use the full meaning of acronyms in the first appearance such as OECD.

Response: Thanks for pointing out this. We have carefully reviewed the entire manuscript to ensure that all acronyms are properly introduced at their first appearance. (p.3, l. 26)

Use the full meaning of an acronyms for a subtopic such as at 2.1

Response: We have provided the full meaning of the acronym as suggested. (p.6, l. 98)

What is homogeneous convenience sampling?

Response: In this revised version, we have clarified this concept and provided a justification for employing this sampling method. Homogeneous convenience sampling refers to the recruitment of accessible individuals who share salient characteristics relevant to the research objectives [57]. In this study, the target population comprised music teachers in Chinese higher education institutions. Restricting recruitment to this group avoided potential heterogeneity associated with including teachers from primary or secondary schools, whose institutional and pedagogical contexts differ substantially. This strategy ensured that participants were situated within similar organisational environments and subject to comparable professional demands. As emphasised by Jager et al. [57], homogeneous convenience sampling improves the representativeness, reduces estimation bias, and increases the generalisability of findings compared with conventional convenience sampling approaches that pool individuals from disparate backgrounds. (p.10, ll. 189-194)

I was initially confused when the authors mentioned that the scales are already validated but later saw that validity evidences where provided as well in the paper. It will be useful to hint the reader that despite that the scales have been validated in previous studies, it will also be validated in the study (page 5).

Response: To avoid potential confusion, we have removed the previous phrasing that may have implied the scales did not require further evaluation. The revised text now clearly indicates that we used well-established scales and conducted standard psychometric assessments (reliability and validity analyses) within our sample, which is routine practice when applying established instruments. (p.11, ll. 222-224)

It could be more convenient for readers to have the diagrams close to where they are discussed instead of having them at the appendix.

Response: According to PLOS ONE submission guidelines, figures must be submitted as separate files rather than embedded within the manuscript text. All figures are properly cited in ascending numerical order in the main text, and the journal’s editorial team will integrate them appropriately in the published version according to standard formatting practices.

Major Comments

A major confusion I have in the study is about the group sampled. At some point teachers in higher education institution was mentioned and at another point in-service teachers were mentioned. Are they the same?

Response: The two terms refer to the same population in our study—music teachers employed at higher education institutions. The term “in-service teachers” was intended to emphasise that participants were actively working teachers (as opposed to pre-service trainee teachers), but this wording introduced unnecessary ambiguity. In this revised version, we have removed this term.

I am worried about the sample size. Only 83 samples was used for the final analysis for such a complex model.

Response: Thanks for this comment. While the final sample included 83 participants, this exceeds the minimum recommended sample size of 69 for detecting medium-sized effects (path coefficient ≥ 0.3) using the inverse square root method [58–61]. This method is conservative and slightly overestimates the sample size needed to achieve a given power [62]. Moreover, in PLS-SEM, the overall complexity of the structural model has little influence on sample size requirements [62]. Therefore, despite the model’s apparent complexity relative to the sample size, 83 participants are adequate for detecting the hypothesised effects. We have clarified this point in the revised manuscript. (p.10, ll. 195-201, p.11, ll. 211-215)

There are no discussion regarding the fit of the model. I think that is an important aspect of a model like this

Response: We agree that model evaluation is essential in SEM research. However, the notion of model fit in covariance-based SEM (CB-SEM) cannot be directly transferred to PLS-SEM due to their fundamentally different estimation logics. Whereas CB-SEM relies on global goodness-of-fit indices derived from discrepancies between empirical and model-implied covariance matrices, PLS-SEM instead emphasises variance explanation and predictive accuracy [62]. Consequently, traditional CB-SEM fit indices (e.g., χ², RMSEA, CFI) are not applicable in PLS-SEM, and the use of global goodness-of-fit measures remains controversial [62,72,96]. Following Hair et al. [62], we therefore did not report global fit indices in the manuscript. Nevertheless, to additionally address your concern, we also reported the model’s high out-of-sample predictive power, confirmed via the PLSpredict procedure (p.17, ll. 295-301). The results demonstrate that endogenous constructs possess robust predictive capabilities, confirming their significantly superior predictive performance compared to simple linear models [96].

Does the authors check if there are connections between the items or simply assumed there is none?

Response: We did not assume that the items were independent. Instead, we carefully examined their relationships to ensure the measurement model was valid. All items loaded strongly on their intended constructs, and cross-loadings analysis showed no substantial loadings on other constructs. Discriminant validity was further checked using the HTMT criterion, with only the BI–UB HTMT (0.94) slightly above 0.90, which is theoretically acceptable given the strong link between behavioural intentions and actual usage. Internal consistency across constructs was excellent (AVE > 0.5, composite reliability > 0.7). Overall, these checks confirm that the item relationships were thoroughly evaluated, supporting a reliable and valid measurement model for our PLS-SEM analysis.

Why not run the same model with the available moderators and compare the fit of the model for the sample in the study. That could present a robust and more informative model that could inform policy.

Response: We are sincerely grateful to the reviewer for such valuable suggestions, which have contributed to enhancing the robustness and informational content of the model. Considering the PLS-SEM methodology employed in this study, we have conducted a systematic multi-group analysis (MGA) to demonstrate the comparisons between the models. This encompasses subgroup analyses of the overall model, such as reporting R² values, alongside examining the variation in path effects across different groups. This analysis examined both path-level differences and variations in overall model explanatory power across subgroups. Consistent with PLS-SEM methods, we focused on predictive performance and explained variance rather than global fit indices, enabling us to gain deeper insights into the model's applicability and boundary conditions by evaluating performance across different demographic subgroups.

The MGA results reveal some significant heterogeneity across demographic groups (see Table 5). While gender demonstrates no significant moderation effects across any pathways, age and experience show substantial differences in key relationships. To fully integrate these rich findings, we have substantially revised the manuscript. The specific changes are as follows:

• The abstract has been revised to incorporate differing findings derived from multiple group analysis.

• We have revised the Hypothesis Development section, removing our previous rationale for excluding moderators and instead introducing the theoretical basis for testing these effects. (p.9, ll. 168-172)

• A new subsection, Multi-Group Analysis, has been added to the findings (pp.17-19, ll. 308-336). This section presents the full findings from the MGA, which are summarised in a new Table 5.

• The Discussion section has been significantly enriched to interpret these moderation findings and explore their practical implications for different teacher cohorts.

• The Limitations section has been updated. We have removed the previous limitation regarding the exclusion of moderators and have instead added a more nuanced discussion on the statistical power of the multigroup analysis.

Reviewer #2

The manuscript was written in a clear and coherent manner, and all the necessary aspects of the study were elucidated concisely. However, there is only one statement that needs the authors to adjust so that it is accurate and meaningful. The authors should add the phrase "and learning" to the end of the third sentence in paragraph 2, under the "Introduction" heading on page 1."

Response: Thanks for your suggestions. We have revised the sentence as requested. (p.3, l. 40)

---

## [Editor Report · Decision Letter 1]

23 Sep 2025

Chinese music teachers value effectiveness, social, and facilitating factors over ease of use in ICT integration: A PLS-SEM study

PONE-D-25-31236R1

Dear Dr. Zhang,

We’re pleased to inform you that your manuscript has been judged scientifically suitable for publication and will be formally accepted for publication once it meets all outstanding technical requirements.

Kind regards,

Musa Adekunle Ayanwale, Ph.D

Academic Editor

PLOS ONE

Additional Editor Comments (optional):

The revision meets expectations.
---

## [Editor Report · Acceptance letter]

PONE-D-25-31236R1

PLOS ONE

Dear Dr. Zhang,

I'm pleased to inform you that your manuscript has been deemed suitable for publication in PLOS ONE. Congratulations! Your manuscript is now being handed over to our production team.

Kind regards,

on behalf of

Dr Musa Adekunle Ayanwale

Academic Editor

PLOS ONE